# Efficient Image Segmentation of Cardiac Conditions after Basketball Using a Deep Neural Network

**Jian Ma [1] and Wenfa Li [2,3,***

1  School of Software Engineering of USTC, University of Science and Technology of China, Suzhou 215123, China
2  The Institute of Artificial Intelligence, University of Science and Technology Beijing, Beijing 100083, China
3  College of Robotics, Beijing Union University, Beijing 100101, China
*  Correspondence: liwenfa@ustb.edu.cn

**Abstract:** The evaluation of heart health status is the reference standard for measuring the intensity of exercise performed by different individuals. Thus, the effective analysis of heart conditions is an important research topic. In this study, we propose a system designed to segment images of the right ventricle. In this system, the right ventricle of the heart is segmented using an improved model called RAU-Net. The sensitivity and specificity of the network are enhanced by improving the loss function. We adopted an extended convolution rather than ordinary convolution to increase the receptive field of the network. In the network-sampling phase, we introduce an attention module to improve the accuracy of network segmentation. In the encoding and decoding stages, we also introduce three residual modules to solve the gradient explosion problem. The results of experiments are provided to show that the proposed algorithm exhibited better segmentation accuracy than an existing algorithm. Moreover, the algorithm can also be trained more rapidly and efficiently.

**Keywords:** deep neural network; attention module; encoding and decoding stage

## 1. Introduction

The popularization of health and fitness training has had a positive impact on the global development and adoption of basketball. Fitness training mainly includes training of muscle strength, sensitivity, and coordination, as well as muscle endurance, cardiopulmonary endurance, neuromuscular components, and it can improve the aforementioned aspects of the body through physical training [1]. College students commonly play basketball in nearly all parts of the world. However, many college students do not pay special attention to their physical fitness and may easily be injured while playing. In this study, we investigated the knowledge of students in seven developing countries about fitness training using a questionnaire survey. In total, 300 questionnaires were sent. Through screening, 289 were effective, with a recovery rate of 96%. Among the 39% of the students who had engaged in fitness training, 30% of them were familiar with the topic and 31% were not. This also shows that the popularization of fitness training in developing countries still involves some notable problems [2].

Basketball is a high-intensity competitive sport that emphasizes physicality and involves considerable requirements in terms of height, body shape, and physical capacities. In terms of physical muscle strength, college students often perform extensive strength training in basketball. Throwing the ball, changing direction, and dunking also require a great deal of flexibility, athleticism, and explosive physical power. Monitoring the cardiac functions of an individual while playing basketball thus has certain significance. Therefore, we can consider that fitness training highlights the importance of heart function [3]. In may developed countries, the mortality rate associated with cardiovascular disease has decreased rapidly in recent years. By contrast, although the mortality rate of cardiovascular disease in many developing countries has improved, it remains much higher than

that in developed countries [4]. The knowledge regarding the shape and function of the heart can help in diagnosis and treatment; however, there is a lack of relevant quantitative information. The use of medical imaging technology to assist clinical diagnosis has thus become particularly important.

## 2. Related Works

At present, cardiac MR imaging is one of the most important, accurate, and noninvasive diagnostic tools for imaging cardiac structure and function. Doctors usually analyze a patient's cardiac MR images and calculate the continuous dynamic changes in the left and right ventricular volumes in the process of contraction and relaxation. This approach can be used to determine parameters, such as cardiac end-diastolic and -systolic volume, stroke output, and ejection fraction, to judge an individual's cardiovascular health status. Therefore, accurate segmentation of ventricles in such imaging is very important. A cardiac MR image is shown in Figure 1.

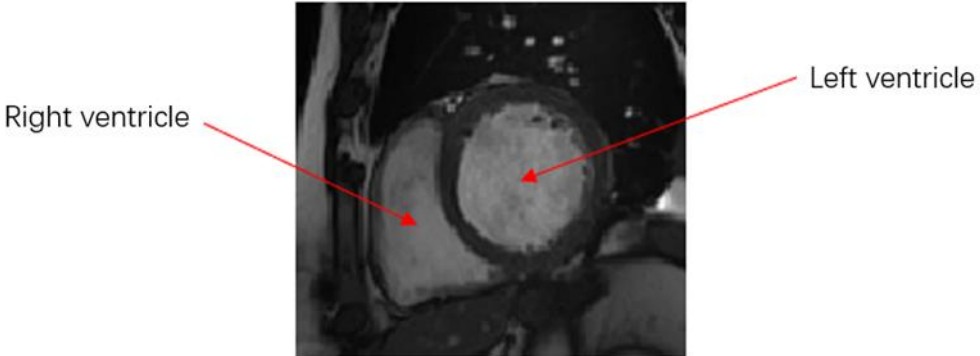

**Figure 1.** MR image of the heart.

Common ventricular segmentation techniques can be roughly divided into four categories, including segmentation algorithms based on thresholds and those based on clustering technology, deformable models, and neural networks.

Segmentation algorithm based on threshold. Threshold segmentation is the most commonly used general image segmentation method. This approach requires the target to be segmented to have special quantitative characteristics. The essence of segmentation is to find pixels within a specified threshold range. The appropriate threshold can be determined manually or by using an algorithm. Ng et al. [5] proposed a medical image segmentation method based on a threshold value, which adopted watershed segmentation and texture-based region merging. Kotropoulos et al. [6] proposed a method to filter an image first, then perform threshold segmentation, and finally set a function in a support-vector machine (SVM) classifier to segment the image. Khare et al. [7] proposed soft threshold segmentation in medicine that used a membership function to classify pixels to achieve image segmentation. This method is highly robust and does not require manual interaction for automatic segmentation.

For segmentation algorithm based on clustering technology. Common clustering techniques and segmentation methods include $k$-means and fuzzy C-means. $k$-means clustering calculates the average number of separated classes or cluster values in an image and takes the average value closest to the image for segmentation. Nandagopalan et al. proposed a segmentation method based on $k$-means to improve segmentation speed. Similarly, Li et al. proposed a volume-based medical image segmentation method, and Kumbhar et al. [8] used a trained $k$-means clustering method for the MR segmentation. Their algorithm minimized the change in a cluster through iteration, allocated pixels that violate a given mark to the cluster, and reassigned pixels until all pixels were divided into corresponding categories. In medical image processing, the FCM algorithm combining C-means and fuzzy theory has shown promising results. The performance of FCM can improve the accuracy of Kumbhar

et al.'s [9] segmentation method by adding spatial influence to the objective function, or using techniques that can transform nonlinear problems into antecedent problems more effectively [10]. For example, Li et al. [11] modified an objective function by adding an influence term defined by the label in the neighborhood pixel. Ozyurt et al. [12] proposed a similar MR segmentation method. To shorten the segmentation time, Szilagyi et al. [13] introduced new factors based on the FCM. Balafar et al. [14] used wavelet transform for gray value processing to reduce the noise in an image before clustering it based on the central grey level.

Segmentation methods based on deformable models. Compared to the two types of segmentation methods discussed above, segmentation methods based on deformable models are more flexible and can be used for complex segmentation. Processes based on segmentation methods that use deformable models can be considered as a model of curve evolution. This is based on the target boundary. The characteristics considered by the target boundary include the shape, smoothness, internal force, and external force acting on the segmented object. All these factors affect the effectiveness of the obtained results. A closed curve and its shape in an image are used to reach the object boundary. The piecewise continuity of these methods is their main advantage. According to their approach to motion contour tracking, variability models can be divided into parametric and geometric deformable models. Parametric deformable models are described by a finite number of parameters, and are also known as active contour methods. The shape model is represented using a generated parameter curve. The parametric deformable model is explicitly tracked by sampling the contour points and tracking the evolution process. This model has the advantages of high computational efficiency and convenience for real-time applications. The development of parametric deformable models is closely related to the snake method [15], which is sensitive to initial conditions. The motion contour can stop at the location where the local function is minimal or at the location where the gradient amplitude is too small and the external force area is zero. To prevent the curve from contracting or stopping at a local minima, Cohen et al.'s [16] method expanded or contracted the contour for weak gradient lengths by adding momentum. A set deformable model is established based on the level-set method [17]. The level set method can easily deal with topological changes, and the geometric properties of the contour can be calculated implicitly, which reduces the computational complexity of the geometric deformation model. Malladi et al. [18] provided an algorithm that used gradient information to define a velocity function and added curvature influence to maintain a smooth contour. When the contour moves to the structural boundary, the increase in the amplitude of the gradient reduces the velocity value, which slows down the evolution of the contour. If noise is present in the image, the contour may be produced in segmentation only after a long processing time. To solve this phenomenon, we apply edge [19] and regional strength terms [20] to improve the model.

Segmentation algorithm based on neural network models. Owing to their unique advantages, deep neural networks play an important role in image processing. Deep learning has thus become the most common approach in medical image segmentation [21]. Because the application of deep learning to the field of medical image segmentation has played a significant role in improving segmentation accuracy, many such algorithms has been proposed. In 1998, the LetNet-5 architecture was proposed [22]. AlexNet [23] and other models [24] won the championship of the ImageNet competition in 2012, and convolutional neural networks were subsequently widely recognized and adopted. U-Net is a well-known representative example [25]. The innovative U-Net model adopted an up-down sampling structure and jump connection. This means that the model only needs to process the image segmentation once to perform image segmentation. Driven by these advances, convolutional neural networks are now of great significance in heart-image segmentation. In clinical diagnosis, heart images must usually be segmented to obtain a heart-related index to assist the diagnosis. Tran et al. [26] used FCN to segment the left and right ventricles from cardiac MR images and demonstrated its advantages in terms

of accuracy and speed. Subsequently, more segmentation methods were developed by Bai et al. [27], particularly a segmentation method based on U-Net [28]. The combination of spatial and temporal background information has also been an important research direction, including segmentation of the heart at the end of diastole and systole. Shape-based constraints have been shown to be effective when segmenting the left ventricle using other types of machine learning methods, and were included in a deep learning strategy based on anatomical constraints [29]. Other important works have considered MR atrial segmentation [30], CT whole-heart segmentation [31], and 3D ultrasound image sequence left ventricular segmentation [32]. Given the many challenges evident in the field of heart image segmentation, considerable room for improvement remains in terms of the performance of heart segmentation methods.

Although there are many types of segmentation algorithms, most methods are based on left ventricular segmentation, owing to the complexity of heart images and differences in heart images among different races. Few studies have been conducted on the segmentation of the right ventricle, and the accuracy of most methods does not suffice. Therefore, we adopted the U-Net network as a basis for further development because it is widely used in right ventricular segmentation.

In this study, we designed and implemented a right ventricular image segmentation system. In this system, the right ventricular heart is segmented according to the improved U-Net, which we refer to as RAU-Net. The relevant cardiac index was calculated based on the segmentation results. The contributions of this study are summarized as follows:

(1)  We propose targeted improvements to address some limitations of traditional U-Net. We adopt extended convolution to increase the receptive field.
(2)  We propose RAU-Net by introducing an attention module and improving the network loss function.
(3)  We introduce residual module into the original network to improve the training speed.

The algorithm not only ensures the training speed, but also further improves the accuracy of network segmentation. Finally, our experimental results verified the effectiveness of the improvement. Thus, the proposed approach can be implemented in clinical practice to perform auxiliary diagnosis.

## 3. Algorithm Design

Through the reproduction of the right ventricle segmented by the basic U-Net network, it may be observed that although good results were achieved in some clearer images, some problems remain with complex images, which considerably affects the accuracy of segmentation. The shortcomings of the model mainly include the following:

(1)  The model does not notice that the receptive field significantly affects the segmentation performance of the model. The size of the receptive field imparts different sensitivity to the target segmentation area. Therefore, a change in the receptive field size leads to a change in the performance of the segmentation model.
(2)  Both over- and under-segmentation are possible. Owing to the large difference in the sample images, the algorithm may segment the segmented and non-segmented regions incorrectly.
(3)  The training speed of the model is slow.

To address these shortcomings, we propose the following improvements:

(1)  An extended convolution is introduced to increase the receptive field size of the network model.
(2)  An attention module is introduced to improve segmentation accuracy.
(3)  A residual module is introduced into the original network to improve the training speed.

### 3.1. Design of Extended Convolution

The network downsampling stage is known as the shrink path, and is used to extract the features of the input image. In this process, two convolution operations with a convolution kernel of $3 \times 3$, padding value of 1, and step size of 1 were performed, and ReLU was selected as the activation function. The receptive field of the two convolution operations is the same as that of a convolution operation with a 5 by 5 kernel, although it reduces the required computation. After two convolutions, the maximum pooling operation is performed on the feature image with a size of $2 \times 2$ and a step size of 2. Thus, a feature image with half the original size is obtained. After shrinking the path, the number of channels in the input stage is set to 64, then the number of channels is doubled after each stage, and the number of channels is repeated four times to 512 channels. After $3 \times 3$ convolution of the feature image, a feature image with 1024 channels is obtained.

Because the output requires an image of the same size as the original image, the expansion path requires a reduction in the number of channels. After a $3 \times 3$ convolution of the feature image, a deconvolution is carried out with a convolution kernel size of $3 \times 3$ and step size of 2, the image size is expanded to the original 2, and the number of channels becomes 512. The image obtained by deconvolution is pieced together with an image of the same size on the shrink path to complete a jump connection. Two convolution operations with a convolution kernel of $3 \times 3$, padding value of 1, and step size of 1 are performed on the feature image obtained, the ReLU activation function is applied, and the number of channels is divided in half. The convolution operation is repeated four times; the size of the feature image is doubled each time, and the number of channels is halved each time. Finally, convolution with a convolution kernel of $1 \times 1$ is performed to reduce the number of feature image channels from 64 to 1 and output feature images.

In the problem of image segmentation, the pooling operation in the process of downsampling leads to a reduction in image resolution and the loss of key information. To solve these problems, we propose extended convolution as a new convolution method. The core idea of extended convolution is to add 0-value pixels between each pixel to increase the size of the nucleus in a disguised manner, without changing the size of the feature map to increase the receptive field. The number of pixels with a value of zero was determined by the expansion parameter. For example, take a convolution kernel with a size of $3 \times 3$. When the expansion parameter is set to 2, a 0-value pixel is added between each row and column of the convolution kernel. The size of its receptive field increases from three to seven. Figure 2 shows the change in the receptive field size of a $3 \times 3$ convolution kernel when the expansion parameters are set to 1, 2, and 4.

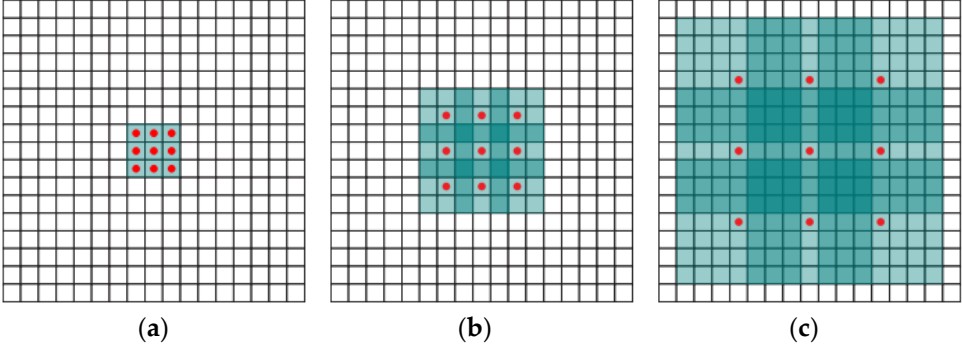

(a)            (b)            (c)

**Figure 2.** Schematic diagram of receptive field size changes under different expansion parameters. (**a**) The expansion parameter is 1. (**b**) The expansion parameter is 2. (**c**) The expansion parameter is 4.

Figure 2a shows that when the expansion parameter is 1, the size of the convolution kernel is 3 × 3, and the size of the receptive field is 3. Figure 2b shows that when the expansion parameter is 2, except for the red dot position in the figure, the other green positions are 0-value pixels. The receptive field was expanded to seven without changing the convolution kernel size to 3 × 3 pixels. Figure 2c shows that when the expansion parameter is 4, the receptive field is also expanded to 15 without changing the size of the convolution kernel. It may be observed that the receptive field sizes of the expanded convolution and ordinary convolution differed in proportion to the number of convolution layers. The size of the expanded convolution receptive field is exponentially related to the number of convolutional layers. The steps used to calculate void convolution are as follows:

$$f = (n - 1) \times (k - 1) + k, \tag{1}$$

where $f$ is the size of the hollow convolution kernel, $n$ is the expansion parameter, and $k$ is the size of the ordinary convolution kernel.

$$RF_{i+1} = RF_i + (f - 1) \times S_i, \tag{2}$$

$$S_i = \prod_{i=1}^{i} Stride_i \tag{3}$$

$RF_{i+1}$ indicates the receptive field of the current layer, $RF_i$ represents the receptive field of the upper layer, and $S_i$ is the product of the steps of all the previous layers.

Table 1 lists the exponential growth process of the receptive field with the number of convolution layers. Layer 6 has an expanded receptive field with a size of 127, based on which the extended convolution is applied to the U-Net. Rapidly increasing the receptive field of the network can not only reduce the network parameters, but also improve the speed of image processing without affecting the size of the feature image.

**Table 1.** Extended convolutional receptive field variation table.

| Number of Layers | Convolution Kernel Size | Step | Coefficient of Expansion | Receptive Field Size |
|---|---|---|---|---|
| 1 | 3 | 1 | 2 | 3 |
| 2 | 3 | 1 | 4 | 7 |
| 3 | 3 | 1 | 8 | 15 |
| 4 | 3 | 1 | 16 | 31 |
| 5 | 3 | 1 | 32 | 63 |
| 6 | 3 | 1 | 64 | 127 |

*3.2. Improved Design of Edge Detail Segmentation*

In the process of segmenting the right ventricle, because most of the cardiac MR images are tissue images around the ventricle, only a small part is the target area of the segmentation task, accounting for only approximately 5% of the total images. The cross-entropy loss function used in the network is a simple binary classification function. The network may mistakenly focus on the image around the ventricle, resulting in the network giving higher weight to the nontarget region and the problem of classification imbalance. The most important step in the process of training the model is the selection of an appropriate loss function. By changing the loss function value, the performance of the network can be predicted, and the weight can be updated in the direction of propagation to make the network gradually reach the optimum. Common loss functions include cross-entropy, Dice, and Focal loss.

Ventricular segmentation is essentially a binary classification problem; the cross-entropy loss function is used to train the neural network. The cross-entropy loss function is defined as follows:

$$L_B = \sum_{i=1}^{M} -(T_{oi}\log(P_{oi}) + T_{bi}\log(P_{bi}), \tag{4}$$

where $P_{oi}$ and $P_{bi}$ represent the probability of pixel I in the segmented and nonsegmented regions, respectively. $T_{oi}$ and $T_{bi}$ indicate whether the pixel is in the corresponding area with 1 and 0, respectively. When $T_{oi}$ is 1, pixel I is in the segmented area, and 0 indicates that it is in the non-segmented area $T_{bi}$, and vice versa.

Based on the cross-entropy loss function, the sensitivity and specificity functions are combined as a loss function of the network. The loss function of sensitivity and specificity is shown in Equation (5).

$$L_S = \left( \frac{\sum_{i=1}^{M} P_{oi} T_{oi}}{\sum_{i=1}^{M} P_{oi} T_{oi} + \sum_{i=1}^{M} P_{bi} T_{oi}} + \frac{\sum_{i=1}^{M} P_{bi} T_{bi}}{\sum_{i=1}^{M} P_{bi} T_{bi} + \sum_{i=1}^{M} P_{oi} T_{bi}} \right), \tag{5}$$

where $P_{oi}$, $P_{bi}$, $T_{oi}$, and $T_{bi}$ have the same meaning as in Equation (4). Sensitivity means that the network mistakenly considers the pixels that should be around the ventricle as the tissue pixels around the ventricle, while specificity means that the tissue pixels that should be around the ventricle are mistakenly considered as the ventricular pixels. The improved loss function is shown as Equation (6).

$$L = \alpha L_B + \beta L_S. \tag{6}$$

Here, $\alpha$ and $\beta$ are parameters that can vary, as can the weights of the two loss functions. In this formula, $\alpha = 0.2$, $\beta = 0.8$. On the one hand, this can reduce the weight of the cross-entropy loss function and suppress classification imbalance. However, it can improve the sensitivity and specificity of the network models.

### 3.3. Attention Module

Enabling the neural network to focus more on the target area can also simplify the process of designing the network structure and algorithm. A simple network structure can be used to solve complex problems. The U-Net network adopts the method of long skip connections, which can be disturbed by useless information and live noise, and reduce the performance of the training process. Therefore, the channel attention module is added in the feature extraction stage to highlight the information of the target area to suppress useless information.

Channel attention refers to the features to which the neural network can pay more attention according to the target. The input of a neural network is represented by a channel, and convolution generates new channels. The contribution of each newly generated channel to the feature information extracted by the neural network differed. By changing the weight ratio between channels, the neural network can pay more attention to channels with a high contribution rate, which can improve the ability of the network to judge characteristic information. The implementation process is shown in (7).

$$F_C = F_1 \otimes F_s = \left[ F_1^1 \cdots\cdots F_1^n \right] \otimes \left[ F_s^1 \cdots\cdots F_s^n \right] = \left[ F_1^1 \cdot F_s^1 \cdots\cdots F_1^n \cdot F_s^n \right], \tag{7}$$

where $F_1$ represents the weight sequence obtained by the channel attention module, $F_1^i$ represents the weight of the $i$-th channel (i = 1, 2, ... , c), $F_s$ represents the output characteristic tensor of the spatial attention module, and $F_s^i$ represents the $i$ channel of the spatial attention module.

The attention module normalizes the data in a batch. Batch normalization normalizes the feature information before convolution. The normalization operation adjusts the input

characteristic information to a normal distribution with a mean value of 0 and a variance of 1 using Equation (8).

$$x_k = \frac{x_k - E(x_k)}{\sqrt{var(x_k)}}, \tag{8}$$

where $x_k$ represents the input characteristic information, $E(x_k)$ represents the mean value of $x_k$, and $\text{var}(x_k)$ represents the variance of the feature. Theoretically, the mean and variance should be the calculated values of all the trained images, which leads to a large amount of calculation. The training set was divided into multiple batches, and the values of each batch were used to replace the values of all data, to reduce the amount of calculation required. The calculation of the mean and variance are shown in Equations (9) and (10).

$$\mu = \frac{1}{m} \sum_{i=1}^{m} x_i, \tag{9}$$

$$\sigma^2 = \frac{1}{m} \sum_{i=1}^{m} (x_i - \mu)^2 \tag{10}$$

However, if all the information is adjusted to a normal distribution, the learning ability of the network is significantly reduced. Therefore, after the normalization operation, we introduce Equation (11).

$$y_k = \gamma_k x_k + \beta_k \tag{11}$$

Using $\gamma$ and $\beta$ as learning parameters, we vary the range of the data. Optimal selection was obtained through parameter learning. Finally, the normalization operation and data distribution adjustment are shown in Equations (12) and (13), respectively.

$$x_i = \frac{x_i - \mu}{\sqrt{\sigma^2}} \tag{12}$$

$$y_i = \gamma x_i + \beta \tag{13}$$

*3.4. Network Training Speed Improvement Design*

In a traditional neural network, ideally, the performance of the model improves with a deeper model. However, an increase in the network-level depth makes training increasingly difficult. The main reason for this is that the training process of the network is a gradient descent process. In the descending process, a gradient that is too small weakens the backpropagation training error signal and causes the gradient to disappear. A gradient explosion caused may also occur. These problems are serious, and can even lead to a failure of the neural network training process.

U-Net network is a network structure which is symmetrical in shape and tiled like the letter "U". The left part of the model consists of a downsampling structure, and the right part of model is an upsampling structure. To address the shortcomings of the traditional U-Net network in segmenting the right ventricle, we improved the conventional design. The improved RAU-Net network structure is shown in Figure 3. The improved network replaces the ordinary convolution with an extended convolution, which avoids the problem of information loss in the pooling process of the ordinary convolution, and can increase the receptive field of the network. The attention module was added in the upper sampling stage to direct the network to pay more attention to the segmentation target area, to solve the confusion between the right ventricle and the surrounding tissue in the segmentation process. Similarly, to solve the problem of characteristic information transmission in the network, we added three residual modules in the encoding and decoding stages. We also designed a deeper structure to address the problem of gradient explosion caused by an excessively deep network. The same training method as that in the previous section was used to train the model, and the loss function in the original network was replaced by the improved loss function.

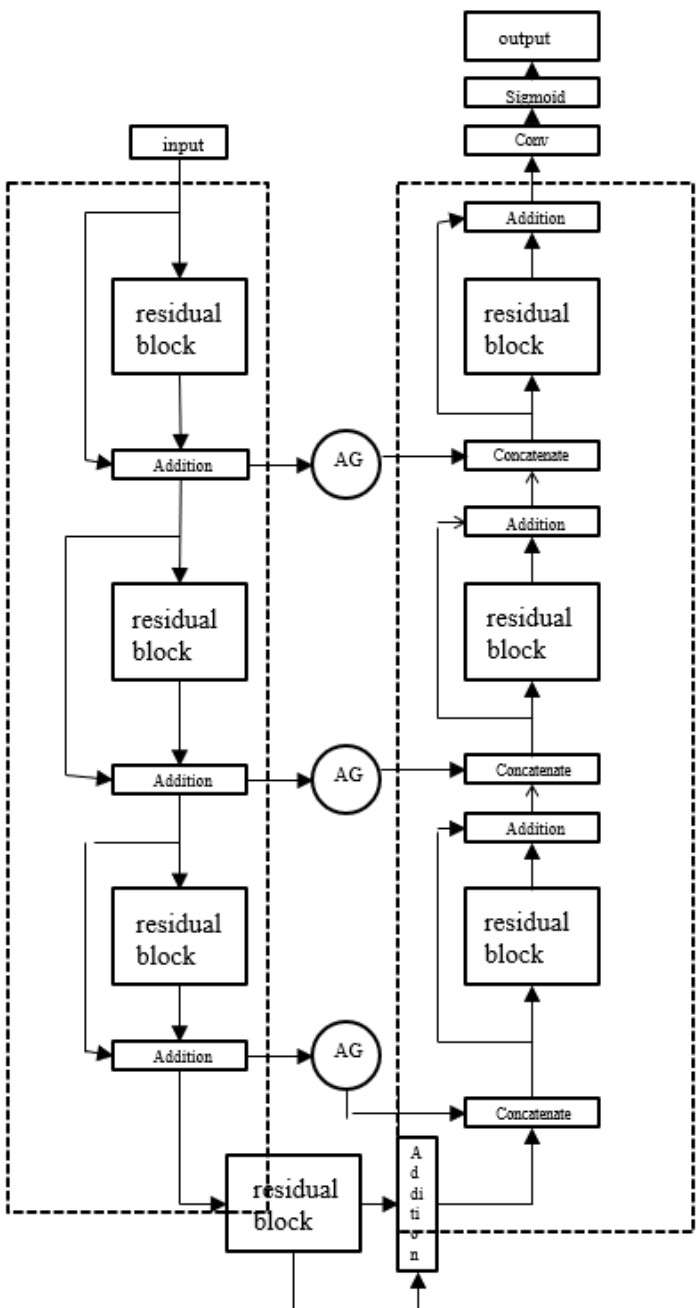

**Figure 3.** RAU-Net network architecture.

## 4. Experiment

We developed a system based on the Electron platform, which is simple to use, quick to start, and can meet the requirements of the system. The software development environment included the Windows 10–64-bit operating system, while the hardware environment included an I7-8750H processor and a GTX1050Ti graphics card.

In this study, we used the Automated Heart Disease Diagnosis Challenge (ADDC) dataset, announced at the 2017 Medical Image Computing and Computer Assisted Intervention (MICCAI) conference. The ADDC dataset consists of cardiac MRI images of 150 subjects taken over 6 years using two different MRI scanners, and is the largest publicly available dataset. According to physiological parameters relating to cardiac health, 150 subjects were evenly divided into five groups, and each group was divided into normal, previous myocardial infarction, right ventricular abnormality, dilated heart disease, and

hypertrophic heart disease. The corresponding physiological parameters of each group were normal, left ventricular ejection fraction less than 40% with abnormal myocardial contraction, right ventricular volume more than 100 mL/m$^2$ right ventricular ejection fraction less than 40%, left ventricular ejection fraction less than 40%, left ventricular volume more than 100 mL/m$^2$, left ventricular volume more than 110 mL/m$^2$, and myocardium segmental diastolic thickness more than 15 m. In addition, the ADDC data also provides ground-truth data that can be used as a training set, as well as 50 samples in a testing that do not include annotated sketched results, and can be used to obtain the network performance by uploading the automatic segmentation results.

The training effectiveness of the datasets plays a significant role in the final performance of the network. Training neural networks generally requires a massive amount of data to achieve the desired results; however, for medical images, the amount of data manually annotated by experts is very small. Inadequate data can cause overfitting. Overfitting leads to excellent performance on the training set, but unsatisfactory performance on the testing set. Therefore, the data were enhanced via techniques, including rotation and mirroring, to expand the data volume and prevent overfitting as shown in Figure 4.

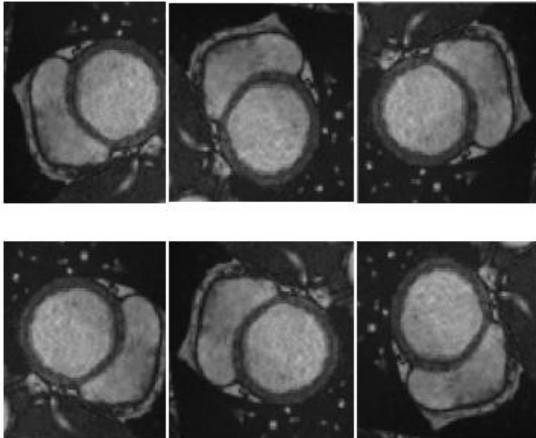

**Figure 4.** Schematic diagram of data enhancement.

The U-Net network was selected as the baseline network for the following reasons.

(1) Deep learning for neural network training requires a large amount of data, while relatively few cardiac MR image datasets with expert labels are available. The U-Net network can also have better segmentation performance with fewer training data.
(2) The U-Net architecture is a simple symmetrical structure, which is convenient for improvement.
(3) U-Net network is widely used in medical image segmentation, which can be easily realized as a benchmark network combined with other algorithms.

Figure 5 shows the change in the loss values of the two networks in the verification set. The red curve represents the loss value change of U-Net on the training set, and the blue curve represents the loss value change of RAU-Net on the training set. It may be observed that the RAU-Net network exhibited a better convergence speed and convergence effect than the U-Net network on the verification set.

The following model parameters were obtained by recording the experimental output results and visualization technology. In terms of the loss value of the training and verification set, the former can judge whether the learning state of the network was normal, and the latter can verify whether the network was effective enough to segment the right ventricle. The Dice coefficient of the training set can be used to judge whether the network is learning, and the change of the similarity between segmentation results and expert annotation results, during training. Similarly, that of the verification set can obtain the ability of the trained network model to segment the right ventricle. When the network

training data loss value converged to 0.4, the network training speed was slow. When the validation set reached a convergence state, the value was 0.84, and the segmentation quality was low.

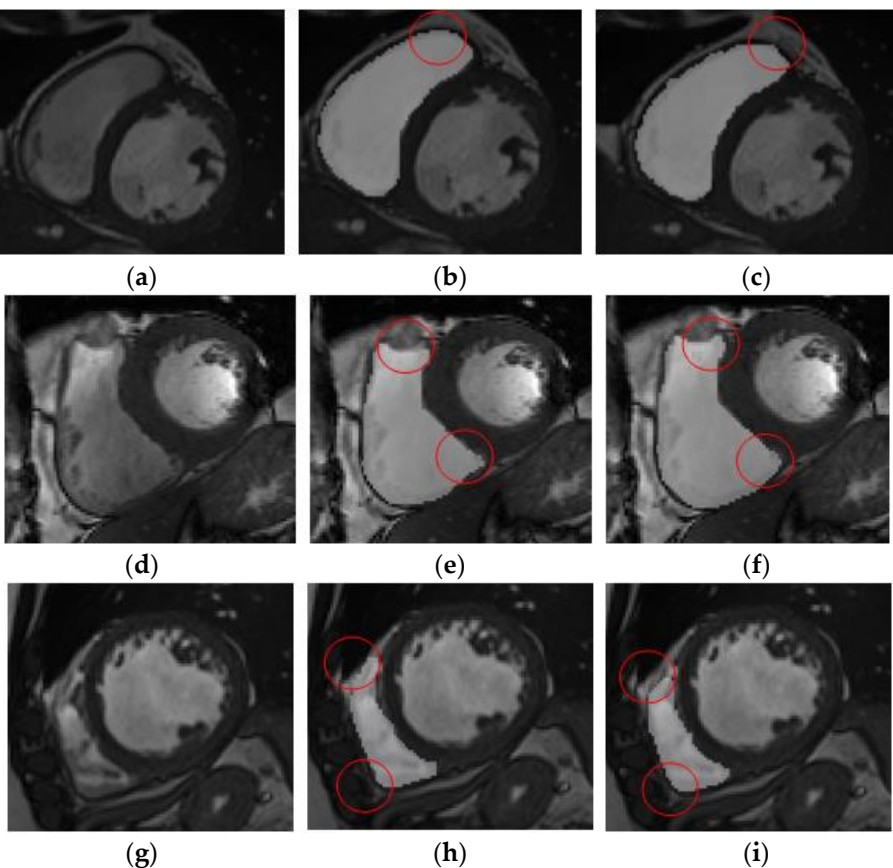

**Figure 5.** Comparison of network segmentation results. (**a**) The input feature image. (**b**) The resulting images of the U-Net network segmentation. (**c**) The gold-standard image marked by experts. (**d**) The input feature image. (**e**) The resulting images of the U-Net network segmentation. (**f**) The gold-standard image marked by experts. (**g**) The input feature image. (**h**) The resulting images of the U-Net network segmentation. (**i**) The gold-standard image marked by experts.

The results of right ventricle segmentation by the U-Net network are shown in Figure 5, which shows three groups of images, each of which is divided into three pieces. Figure 5a,d,g shows the input feature image, Figure 5b,e,h, which are the resulting images of the U-Net network segmentation, and Figure 5c,f,i is the gold-standard image marked by experts.

It may be observed from Figure 5b,c that the result of network segmentation was consistent with the result image marked by experts. However, it may be observed from Figure 5e,f that the network-segmented and expert-labeled images were under-segmented at the apex of the heart. It may be observed from Figure 5h,i that the network-segmented image and expert-labeled image exhibit the problem of over-segmentation near the ventricular wall. The position shown in the red circle represents the difference between the gold-standard image and the segmentation prediction area. Therefore, although the results of the U-Net network were feasible for segmentation of the right ventricle, there was an error in the segmentation of images with complex ventricular shapes.

Figure 6 shows the changes in the loss values of the two networks in the verification set. The red curve represents the loss value change of U-Net on the training set, and the blue curve represents the loss value change of RAU-Net on the training set. It may be observed that although the convergence speed and convergence effect of RAU-Net network

on the verification set were better than those of U-Net network, there was still a loss value of about 0.2. This shows that the network segmentation performance still did not reach the expected effect.

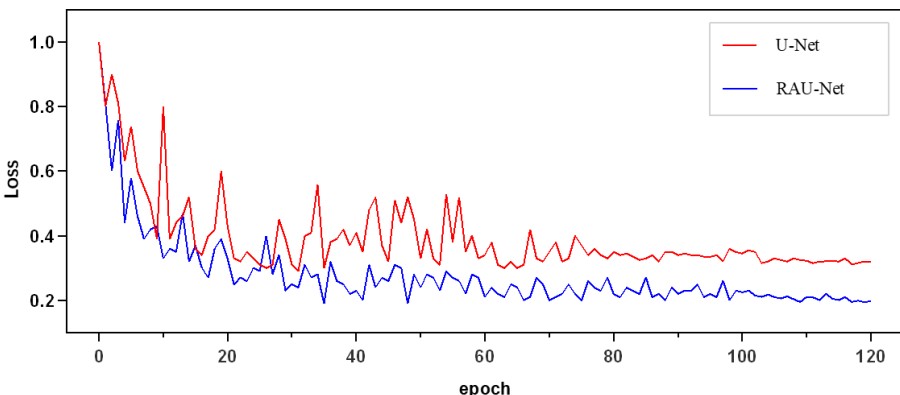

**Figure 6.** Variation in the loss value on the verification set of two networks.

Figure 7 shows the change in the Dice coefficients of the two networks on the verification set. The red curve represents the change in the Dice coefficient of U-Net on the verification set, and the blue curve represents the change in the Dice coefficient of RAU-Net on the verification set. It may be observed that the fitting effect of the RAU-Net network was better than that of the U-Net network, but the Dice coefficient was 0.91 after dividing 120 batches, indicating that the performance of the network still has room for improvement.

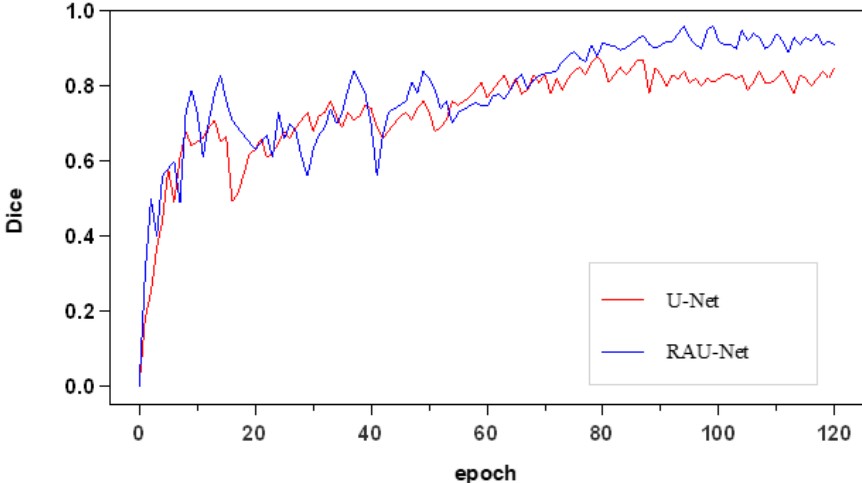

**Figure 7.** Dice coefficient variation of two networks in the validation set.

The results of right ventricle segmentation by the RAU-Net network are shown in Figure 8, which shows two groups of images, each of which is divided into three groups. Figure 8a,d are the input feature images, Figure 8b,e are the resulting images of the U-Net network segmentation, and Figure 8c,f are the gold-standard images marked by experts. The position shown in the red circle represents the difference between the gold-standard image and the segmentation prediction area.

Table 2 compares the algorithm in this study with the winning algorithm in the MICCAI challenge to verify that the algorithm in this study exhibited excellent performance in segmenting images.

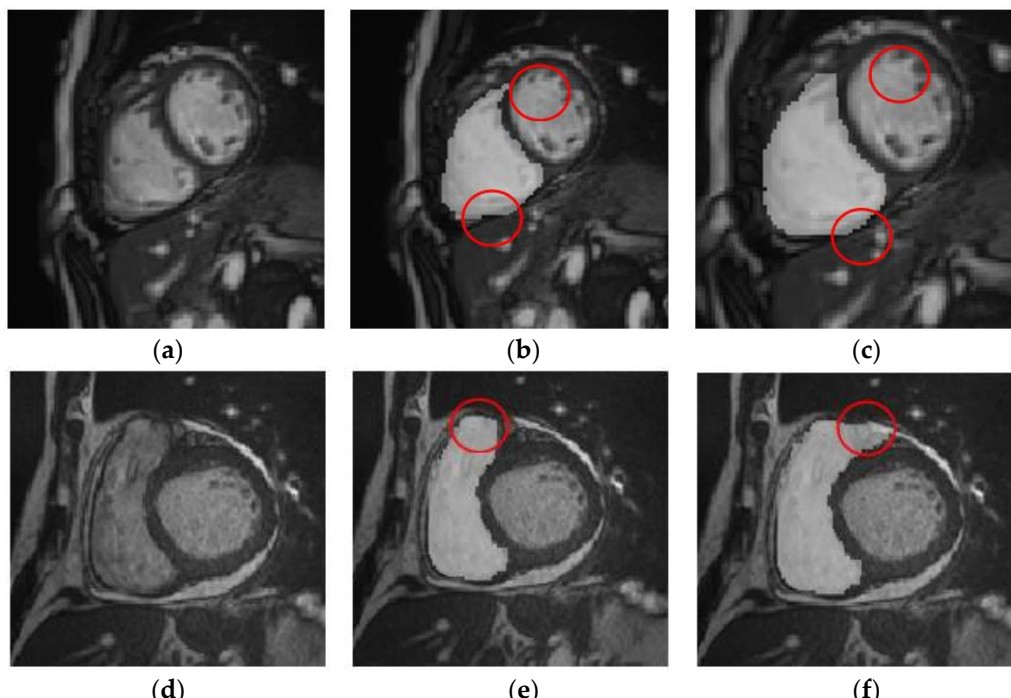

**Figure 8.** Comparison of network segmentation results. (**a**) Input feature images 1. (**b**) Resulting images of the U-Net network segmentation 1. (**c**) Gold-standard images marked by experts. (**d**) Input feature images 2. (**e**) Resulting images of the U-Net network segmentation 2. (**f**) Gold-standard images marked by experts.

**Table 2.** Comparison between the proposed algorithm and other algorithms.

| Method | Loss | Dice |
|---|---|---|
| Szilagyi et al. [14] | 0.31 | 0.89 |
| Siddiqi et al. [21] | 0.22 | 0.91 |
| Bai et al. [27] | 0.14 | 0.93 |
| Ours | 0.09 | 0.95 |

The results of the experiments showed that the improved algorithm enhanced the sensitivity and specificity of the network to some extent. The use of extended convolution instead of ordinary convolution to increase the receptive field of the network also played a vital role. An attention module was introduced in the network sampling stage to further improve the accuracy of network segmentation. In the encoding and decoding stages, three residual modules are added to solve the problem of gradient explosion and improve the training speed of the network.

## 5. Conclusions

In this study, we analyzed the health of the heart after the individual played basketball to design a new algorithm. By analyzing the defects of the traditional U-Net network in segmenting the right ventricle, the algorithm improved the U-Net network to form RAU-Net. We improved the network loss function and the sensitivity and specificity of the network. We replaced ordinary convolution with an extended convolution to increase the network's receptive field. In the upsampling stage, we introduced the attention module to confuse the right ventricle with other surrounding tissues in the segmentation process to improve the accuracy of network segmentation. In the encoding and decoding stages, three residual modules were added to solve the issue of gradient explosion in the process of feature-image layer-by-layer transmission in the network and to improve the network-training speed. Finally, we implemented the RAU-Net architecture. The experimental results confirmed that the speed of the model was significantly improved. In the experiment,

although the accuracy of the algorithm was improved, considerable room remained for further improvement, and further research on this subject seems worthwhile.

In subsequent research, our team reflected on these results. For example, to extract more image feature information in the downsampling stage, two identical RAU-Nets were connected, i.e., LRAU-Net, to form a multi-pair encoding–decoding structure. Because the network hierarchy after connection becomes deeper, the network training speed may be reduced, and thus the residual module needs to be further improved. We plan to further consider equipping the algorithm with a corresponding system, analyzing the system requirements based on clinicians' work requirements, and dividing the system into different modules to achieve different functions.

**Author Contributions:** Conceptualization, methodology, J.M. and W.L.; software, J.M.; validation, J.M.; formal analysis, J.M.; investigation, J.M.; resources, J.M.; data curation, W.L.; writing—original draft preparation, J.M.; writing—review and editing, W.L.; visualization, J.M.; supervision, W.L.; project administration, J.M. and W.L.; funding acquisition, J.M. and W.L. All authors have read and agreed to the published version of the manuscript.

**Funding:** This research was supported by the National Natural Science Foundation of China (No. 61972040).

**Data Availability Statement:** The data used to support the findings of this study are available from the corresponding author upon request.

**Conflicts of Interest:** The authors declare that there is no conflict of interest regarding the publication of this paper.

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
