# Peer review of "Efficient Image Segmentation of Cardiac Conditions after Basketball Using a Deep Neural Network"

_electronics, doi:10.3390/electronics12020466_

Round 1
Reviewer 1 Report
In this paper, an improved U-Net algorithm is presented to divide the right ventricle which is named RAU-Net. The performance of the U-Net suffered from two major problems, namely the lose of the Information and the confusion between the right ventricle and its surrounding tissue. This paper improves the network loss function, uses extended convolution and attention module to solve these two problems, which is an interesting study. There are some comments that may help improve the quality of the paper.
a) What is the difference between the RAU-Net algorithm presented in figure 6 and 7 and the RAU-Net algorithm presented in Table 2? What have you done to help get the result from figure 6 and 7 to Table 2? It needs to explain in detail before your Table 2.
b) Can you tell why (a)(d) is better than (b)(e) in Figure 8 to help the readers get your improvements more intuitively?
c) From the introduction to the algorithm, there is an optimization objective to minimize lose and increase dice, which should be relative to the training set data. So from all experiments, it shows the effect of the training set. Then I want to know the effect of the constructed RAU-Unet model in the test set. Please introduce the training set data and test set data used in the experiment in detail.
d) There are some syntax errors, and the English language expression of this paper should be professionally revised.
Author Response
Reply to Reviewer 1:
- The differences between different algorithms proposed by reviewers are explained here. The algorithms in FIG. 6 and FIG. 7 have been given specific networks in the previous chapter of the experiment. Specific differences include extended convolution and improvement of residual network. Extending convolution instead of ordinary convolution plays an important role in increasing the accepted domain of the network. In the codec stage, three residual modules are added to solve the problem of gradient explosion caused by layer by layer transmission of feature images in the network, and improve the training speed of the network.
- This article explains the differences between the improved images proposed by reviewers. As can be seen from Figure (a) and (d), the result of network segmentation is basically consistent with the expert annotated image, that is to say, the segmentation result can better distinguish the position of the left and right valves. However, as can be seen from Figure (b) and Figure (e), there is an undersegmentation problem between the network segmentation image and the expert marked image at the heart uspid valve, resulting in segmentation errors.
- For reviewers who want to have a deeper understanding of the data set. Here is a detailed introduction of the data set. In this paper, the Automated Heart Disease Diagnosis Challenge announced at the 2017 Medical Image Computing and Computer Assisted Intervention (MIC-CAI). Conference was used Automated Cardiac Diagnosis Challenge (ACDC) dataset. The ACDC dataset consists of cardiac MRI images of 150 subjects taken over 6 years using two different MRI scanners and is the largest publicly available dataset known. According to cardiac physiological parameters, 150 subjects were evenly divided into 5 groups, and each group was divided into normal, old myocardial infarction, right ventricular abnormality, dilated heart disease and hypertrophic heart disease. The corresponding physiological parameters of each group were normal, left ventricular ejection fraction less than 40% with abnormal myocardial contraction, right ventricular volume more than 100ml/m2 right ventricular ejection fraction less than 40%, left ventricular ejection fraction less than 40% left ventricular volume more than 100ml/m2, left ventricular volume more than 110ml/m2 and myocardium segmental diastolic thickness more than 15m M. In addition, the ACDC data also provides Ground Truth data that can be used as a training set, and 50 test sets that do not include manually sketched results can be used to upload their automatic segmentation results to obtain network seg-mentation performance.
Reviewer 2 Report
This study intends to enhance the sensitivity and specificity of the U-Net deep neural network using an extended convolution (called RAU-Net) to improve its loss function. The accuracy of network segmentation gets improved in its accuracy from the sampling stage and attention module. The problem of gradient explosion gets resolved after introducing three residual modules in the encoding and decoding stage. The training speed of the network gets improved as well.
"In terms of physical muscle strength, college basketball uses a lot of physical muscle strength." This sentence needs revision. Most sentences in the first paragraph of the Introduction are not easy to read since "in terms of" and "Therefore" appeared a lot of times. The importance of studying the basketball players' cardiac condition has been emphasized, and the segmentation techniques are critical to determine if players' cardiac condition has problems. Several segmentation methods were introduced in the rest of the paragraphs in Introduction section. However, "the segmentation method based on deformable model is more flexible..." this sentence provides a hint that this study is going to adopt this method or not. The entire paragraph explaining this method is hard to read as well. Careful proof reading is required. Citations or empirical studies shall be provided. Then, the next paragraph "Due to the unique advantages of deep neural network..." is easy to read and has well linked to the segmentation method.
The rest of the manuscript providing details about the proposed algorithms and findings which are compared with current literature is fine with me. Figure 3 needs revision. At the end I only find Conclusions but no discussion.
Author Response
Reply to Reviewer 2:
- According to the grammar problems raised by reviewer, the article has been adjusted to reduce the frequency of words such as "in terms of" and delete some unnecessary content.
- In view of the deviation between the introduction and the conclusion proposed by the reviewer, the corresponding content was rewritten.
Reviewer 3 Report
Dear authors, thank you very much for your work. Some of my comments can be found below:
The English in your paper is very poor: please try to improve it comprehensively. Some sentences are incomplete, others are not written in the spirit of English, and so on.
The first section (introduction) should be divided into two sections: an introduction and a related paper.
There is a serious lack of technical data: There is no information about the complexity of the calculations, the type of hardware used, the duration of the calculation process, etc.
Please add future work in more detail: indicate any upcoming projects and ideas you have on this topic.
In my opinion, this is by no means a new scientific contribution, but rather a set of existing and well-known methods applied to a very interesting problem. I therefore suggest significant changes in this direction.
Author Response
Reply to Reviewer 3:
- Made adjustments to the grammatical problems raised by reviewer.
- The introduction proposed by reviewer has been rewritten, which is divided into two parts: "Introduction" and "Relevant papers".
- Some new content has been added for reference to the description of data sets and other problems proposed by reviewer.
- In view of the contribution of the reviewer, I rewrote the conclusion.
Round 2
Reviewer 1 Report
The paper can be accepted now.
Author Response
Thank you for your recognition and correction of the article
Reviewer 2 Report
Figure 3 is still vague to me. More discussions can be provided at the end of this manuscript. Overall, I think this study is still interesting to me.
Author Response
Thank you for your correction of the article, the following is the accurate reply to the comments, please review:
- An explanation has been added at the bottom of Figure 3. Thank you for your comments.
- The explanation for “more discussions” has been added at the end of the manuscript. We will further consider to equip the algorithm with a corresponding system, and analyze the system requirements based on the work needs of doctors, and divide the system into different modules to achieve different functions.
Reviewer 3 Report
The authors have only slightly modified the paper, but it still has some serious flaws (as described in my earlier report).
Unfortunately, I am of the opinion that this work cannot be classified as a valuable scientific contribution.
Author Response
Thank you for your correction of the article, the following is the accurate reply to the comments, please review:
- The first part of the paper has been adjusted, and the “introduction” and “related papers” are described in two parts. Thank you for your comments.
- More details has been added to the experimental part. Includes the type of hardware used, the time complexity, and the introduction of the data set.
The system is built on the electron platform, the platform is simple to develop, quick to get started, can meet the requirements of the system. Development software environment for Windows 10-64-bit operating system, hardware environment for: I7-8750H, GTX1050Ti graphics card. Obviously, the time complexity of this algorithm is O(n). The main direction of this research is to improve the recognition accuracy. In terms of detection speed, it is almost the same as other comparison algorithms mentioned in this paper. In this paper, we used the Automated Heart Disease Diagnosis Challenge (ADDC) dataset announced at the 2017 Medical Image Computing and Computer Assisted Intervention (MICCAI) conference. The ADDC dataset consists of cardiac MRI images of 150 subjects taken over 6 years using two different MRI scanners and is the largest publicly available dataset known.
- It has increased our planning ideas for the future of the project. We will further consider to equip the algorithm with a corresponding system, and analyze the system requirements based on the work needs of doctors, and divide the system into different modules to achieve different functions.
- At the end of the paper, some contributions made by this paper are emphasized. Our team can realize that our contribution to this research is not groundbreaking, but the exploration of the application value of this paper may become the target of our team's future research.
In the following research, our team carried out more thinking. For example, in order to extract more image feature information in the down-sampling stage, two identical RAU-Nets are connected, namely LRAU-Net, to form a multi-pair encoding decoding structure. Because the network hierarchy after connection becomes deeper, the network training speed may be reduced, so the residual module needs to be further improved.
Round 3
Reviewer 3 Report
Dear authors, thank you for the new version of your work.
Once again, the English in your paper is very poor (you can only read the abstract to understand what I mean).
Also, I still think that this is not a new scientific paper at all, but rather a set of existing and well-known methods applied to a very interesting problem.
Author Response
Reply to Reviewer :
First of all, thank you for your suggestion again. Secondly, we systematically revised the language and content of the whole article. The details are as follows:
- The summary has been revised to highlight the contributions and innovations of this paper.
“The evaluation of heart health status is the reference standard for measuring the intensity of exercise performed by different individuals. Thus, the effective analysis of heart conditions is an important research topic. In this study, we propose a system designed to segment images of the right ventricle. In this system, the right ventricle of the heart is segmented using an improved model called RAU-Net. The sensitivity and specificity of the network are enhanced by improving the loss function. We adopted an extended convolution rather than ordinary convolution to increase the receptive field of the network. In the network-sampling phase, we introduce an attention module to improve the accuracy of network segmentation. In the encoding and decoding stages, we also introduce three residual modules to solve the gradient explosion problem. The results of experiments are provided to show that the proposed algorithm exhibited better segmentation accuracy than an existing algorithm. Moreover, the algorithm can also be trained more rapidly and efficiently.”
- The contents of "Introduction" and "Relevant Methods" have been modified. It reduces some unnecessary descriptions that have little to do with this article. In order to improve the integrity of the cohesion between paragraphs, different kinds of segmentation methods are strictly classified and discussed. The text is bold at the beginning of each method paragraph.
“Common ventricular segmentation techniques can be roughly divided into four categories, including segmentation algorithms based on thresholds and those based on clustering technology, deformable models, and neural networks.”(After this paragraph, based on the above four types of methods, the relevant research in recent years is introduced and discussed in detail.)
- 3. Rewritten the content of the article. The modification has been marked with yellow.
Thank you again for your review.